# Recovery of Potential Starter Cultures and Probiotics from Fermented Sorghum (Ting) Slurries

**DOI:** 10.3390/microorganisms11030715

**Published:** 2023-03-09

**Authors:** Seth Molamu Rapoo, Phumudzo Budeli, Mathoto Lydia Thaoge

**Affiliations:** Department of Biotechnology and Food Technology, Tshwane University of Technology, Private Bag X680, Pretoria 0001, South Africabphumu@gmail.com (P.B.)

**Keywords:** fermentation, sorghum, ting, starter cultures, lactic acid bacteria, probiotics, antibacterial activity

## Abstract

Fermented foods are thought to provide a source of probiotics that promote gut health. Consequently, isolation and characterization of fermented food strains and their applications in a controlled fermentation process or as probiotics present a new facet in this area of research. Therefore, the current study sought to identify dominant strains in sorghum-fermented foods (ting) and characterize their probiotic potential in vitro. Recovered isolates were identified as *Lactobacillus helveticus*, *Lactobacillus amylolyticus*, *Lacticaseibacillus paracasei*, *Lacticaseibacillus paracasei* subsp *paracasei*, *Lactiplantibacillus plantarum*, *Levilactobacillus brevis*, *Loigolactobacillus coryniformis* and *Loigolactobacillus coryniformis* subsp *torquens* based on the their 16S rRNA sequences. Increased biomass was noted in seven out of nine under a low pH of 3 and a high bile concentration of 2% in vitro. Bactericidal activities of isolated LABs presented varying degrees of resistance against selected pathogenic bacteria ranging between (1.57 to 41 mm), (10 to 41 mm), and (11.26 to 42 mm) for *Salmonella typhimurium ATTC* 14028, *Staphylococcus aureus* ATTC 6538 and *Escherichia coli ATTC8739*, respectively. Ampicillin, erythromycin, mupirocin, tetracycline and chloramphenicol were able to inhibit growth of all selected LABs. Thus, isolates recovered from ting partially satisfy the potential candidacy for probiotics by virtue of being more tolerant to acid and bile, antibacterial activity and antibiotic resistance.

## 1. Introduction

Fermentation-based food products remain a significant part of our regular diet and are estimated to account for a third of world food supplies. These food products have been reported to be beneficial in alleviation of metabolic diseases, antimicrobial activities, probiotic, cholesterol-lowering characteristics, in addition to serving as an alternative source for bioactive compounds [1,2]. Factors such as rapid urbanization, population growth and cereal importation costs have increased demand for high quality functional foods such as ting [3]. However, sorghum-based food products have not been fully explored and capitalized on. Sorgum is the third most consumed cereal in Africa, preceded by maize and rice [4]; despite that, Sorgum trails behind the aforementioned cereals in terms of industrialization. In Southern Africa, the isolation and characterization of LAB (Lactic acid bacteria) as probiotics has only been isolated from maize-based fermented beverage known as Mageu [5]. As a drought tolerant and a climate-smart crop under the prevailing realities of climate change, the utilization of sorghum is spread across diverse industries [6].

Ting is a traditional fermented sorghum product, which originates from Botswana and in South Africa is consumed by Botswana people throughout the year. The method of making ting is very similar throughout the countries which utilize the natural microflora in sorghum and the process of making the slurry involves mixing the sorghum flour with warm water and allowing it to naturally ferment for 2–3 days at room temperature [7]. The sorghum slurry is then cooked to make ting/bogobe [7]. This type of porridge is commonly consumed in southern African countries as a thin gruel with sugar for breakfast or as a thicker porridge with relish (meat or vegetables or beans) during lunch or supper. It is also considered a popular delicacy during weddings, funerals and traditional functions [7]. Drawbacks associated with occasional fermentation failure, slow fermentation rates, disparities in quality and less acidification have led to better craftsmanship and inception of back-slopping. These processes involve re-inoculating previous successful fermentation microorganisms into a new batch in order to optimize or improve the overall fermentation process [8]. Consequently, there is a renewed interest tailored to identification and characterization of strains with shortened exponential phases to increase fermentation rates in controlled fermentation processes [9,10]. Dominant strains during fermentation of some fermented sorghum-derived foods have been isolated, preserved and utilized for the improvement of other fermentation food-based products [11,12,13]. Some have also been reported to possess probiotic attributes [14].

In modern medicine, antibiotics are used extensively for treatment of numerous bacterial infections in clinical practices. However, an ever increasing body of literature citing bacterial resistance against commonly prescribed antibiotics has sparked public concern [15,16]. Microorganisms should possess inhibitory substances associated with antimicrobial activities and possess resistance to acidic conditions to be considered a probiotic, thus providing a suited alternative [17,18]. In addition, probiotics should be able to withstand fluctuation of bile concentration and pH in the GIT during food digestion to confer health benefits to the host [19]. Over the years, lactic acid bacteria (LABs) have been isolated from a plethora of fermentation-based food products and shown to possess probiotic characteristics [20,21]. However, there is dearth of knowledge relating to identification, isolation and characterization of LABs implicated in fermentation of ting. There is no record of potential probiotics isolated from ting in the literature to the best of the authors’ knowledge to date. Previous studies focused mainly on chemical characterization and microbial diversity of sorghum slurries which were laboratory based [7,22]. Thus, the objectives of the current study were to isolate, identify and screen the probiotic potential of LABs from household-fermented ting slurry.

## 2. Materials and Methods

### 2.1. Sampling, Isolation and Identification of LABs

Fermented uncooked ting slurries were collected from six different locations in Gauteng province (South Africa). The sample sources included Klipgat, Soshanguve near Pretoria city, Tembisa near Johannesburg and two sources North of Pretoria. The sampling locations were chosen in order to obtain diverse species of LABs as sometimes the preparation of ting varies between households; some use vinegar in the preparation and others do not. The samples were collected aseptically in sterile 1 L Schott bottles carried in a cooler box. The samples were stored at 4 °C during the time of the study at the Department of Biotechnology and Food technology. Ting samples were homogenized and ten-fold dilutions were performed. Furthermore, aliquots of 0.1 mL from the prepared serial dilutions were aseptically transferred and spread evenly onto the sterile MRS agar (Merck, Kenilworth, NJ, USA) plates. The plates were incubated at 37 °C for 48 h. The isolates were purified using repetitive streaking on MRS agar. Pure isolates were examined using light microscopy to score cell morphology, motility and Gram’s stain. Catalase and oxidase activity of the isolates were tested using 3% hydrogen peroxide (Merck) and oxidase test strips (Tetramethyl-p-phenylenediamine) (Merck), respectively. Gram-positive, catalase-negative, oxidase negative, non-motile cells were presumptively identified as LAB.

### 2.2. Molecular Characterisation of Selected LABs

LAB isolates (60) were identified at a molecular level using 16S rRNA and pheynylalanyl tRNA synthetase ulpha subunit (*pheS*) gene sequencing by following the method described as previously described [5]. Briefly, colonies were suspended in 1 × TE buffer (pH 8.0), then mixed well to create a homogeneous suspension of cells and bacterial DNA was extracted using a DNA extraction kit (Epicenter, San Antonio, TX, USA) according to the instructions provided by the manufacturer. Thereafter, the integrity and purity of the eluted bacterial DNA were assessed on the 0.8% agarose gel and measured using a Nanodrop spectrophotometer (Nanodrop 2000, Thermo Scientific, Yokohama, Japan). The 16S rRNA gene was amplified from the DNA extracts using polymerase chain reaction (PCR). The 16S rRNA and *phes* gene were amplified using the primers illustrated in Table 1. The following volumes were measured in order to perform the PCR for the 16S rRNA and *pheS* gene: 12.5 μL Master mix (KAPA Biosystems), 1 μL forward primer and 1μL Reverse primer, 9.5 μL RNase nuclease-free water and 1 μL DNA template to make a reaction volume of 25 μL. The PCR conditions for 16S rRNA were as follows: initial denaturation (94 °C for 15 min), 30 cycles of denaturation (95 °C for 1 min), annealing (50 °C for 40 s), extension (72 °C for 1 min 30 s) and final extension (72 °C for 5 min). The PCR conditions for *pheS* gene were as follows: initial denaturation (95 °C for 5 min); 35 cycles of denaturation (95 °C for 1 min), annealing (56 °C for 30 s), extension (72 °C for 1 min 15 s) and final extension (72 °C for 7.0 min). The PCR amplicons were viewed on a 1 × TAE gel stained with 0.75 µL ethidium bromide (120 V run for 40 min). The PCR fragments were visualized under UV light transilluminator. The PCR products were purified using a QIAquick Purification Kit (Southern Cross Biotechnology, Cape Town, South Africa).

### 2.3. 16S rRNA and pheS Sequencing and Phylogenetic Analysis

DNA Sequencing of 60 PCR products were performed at Inqaba Biotech, Pretoria, South Africa using an ABI 3130 genetic analyzer with a Big Dye Terminator version 3.1 cycle sequencing kit. Sequencing of the 60 PCR products were outsourced to INQABA Biotech, South Africa and 16S rRNA and *pheS* gene sequences obtained and were compared with available sequences using NCBI-BLAST Basic Local Alignment Search Tool (BLASTIN programme) National Centre for Biotechnology Information. A phylogenetic tree was constructed using MEGA X, see Appendix A.

### 2.4. Antimicrobial Activity against Clinically Important Food Pathogens and Antibiotics

Antibacterial activity of LAB strains was carried out using the Modified agar overlay method. The LAB were tested against *Salmonella typhimurium* ATCC 14028, *Escherichia coli* ATCC 8739 and *Staphylococcus aureus* ATCC 65838. Briefly, a circle with 3 cm diameter was inoculated with an overnight Lactobacillus strain; MRS agar plates were incubated anaerobically at 37 °C for 48 h. A volume of 5 mL of nutrient agar (Merck) was poured onto the MRS agar (Merck) culture and was left to solidify at room temperature for about 5 min. The pathogens were diluted to make 10-fold dilutions using phosphate buffered saline. A volume of 100 µL of the diluted pathogen was spread onto the 5 mL of an overlaid nutrient agar. The plates were kept at 4 °C for 2 h prior to aerobic incubation for 24 h at 37 °C. Antibiotic resistance was measured using the disc diffusion method. The overnight grown LAB cultures were spread onto MRS agar plates. Diffusion discs impregnated with specific antibiotics were introduced to the inoculated MRS agar plates using a disc dispenser (Oxoid). The plates were incubated anaerobically at 37 °C for 48–72 h. The clear zones around the discs indicating that the LAB was killed by the antibiotic. The clear zones were measured in millimeters using a digital caliper. All experiments were conducted in triplicate.

### 2.5. Bile Tolerance Test

LAB strains were further subjected to bile tolerance. Briefly, bovine bile salt (Sigma, St. Louis, MO, USA) was added to two MRS broth (Merck) volumes to obtain the final concentrations of 0.3% (*w*/*v*) and 2% (*w*/*v*) separately. The MRS broth (Merck) with the added salt was then distributed into test tubes in 9 mL volumes each and autoclaved at 121 °C for 15 min after which it was cooled at 37 °C. The cooled sterilized tubes were then inoculated with 100 µL of overnight bacterial culture of known cell concentration (10^6^ cfu/mL). The MRS broth culture (Merck) without bile salt was used as a control. The test tubes were incubated at 37 °C in a water bath (GFLD83, Berlin, Germany). The bacterial count was periodically determined at initial stage, after 5 h and finally after 24 h by withdrawing 1 mL from each test tube, to conduct a 10-fold serial dilution then spread onto MRS agar plates. The plates were incubated anaerobically at 37 °C for 48 h. The results were recorded as log counts cfu/mL. All experiments were conducted in triplicate.

### 2.6. Tolerance to pH Fluctuation

An acid tolerance test was conducted by sub-culturing LAB strains into MRS broth (Merck) and incubated at 37 °C for 14–16 h. The broth cultures were then centrifuged at 3500 rpm for 10 min at 4 °C to harvest the cells. The resulting pellet was washed twice using phosphate buffered saline (PBS) then re-suspended in sterile MRS broth. About 4% (*v*/*v*) of the MRS suspension was transferred into 10 mL test tubes of sterile phosphate buffered saline (PBS) at pH 2.0, 3.0, 4.0 and 5.0. The tubes were incubated at 37 °C in a water bath. Bacterial cell counts were determined at initial stage and after 1.0, 2.0, 3.0, and 24 h by employing the spread plate technique. The results were recorded as log counts cfu/mL.

### 2.7. Statistical Analysis

Treatments and analyses were carried out in duplicates and the results are presented as mean ± standard deviation. The means were compared at *p* < 0.05 significance level using analysis of variance (ANOVA) (Statgraphics Centurion XVI, version 16.1.11).

## 3. Results

### 3.1. Microbiological and Phenotypic Characteristics of Isolated LABs

Strains isolated from the sorghum slurry samples were characterized using the morphological and biochemical characteristics (Gram stain, oxidase and catalase tests) and are shown in Table 2. From the observed results, a total of 60 isolates were presumptively categorized as LAB. All the 60 isolates were Gram-positive, catalase and oxidase negative.

The results of the PCR products of the amplified isolates (60) are shown on the gel electrophoresis results in Appendix A (Appendix A). All the 60 LAB Isolates yielded DNA fragments of 1466 bp, which is the expected size for 16s rRNA. In this work, the 16S rRNA molecular makers were successfully used for the identification of the LAB isolates at a species level. All the 60 samples were positive for 16S rRNA. The identified LAB belonged to the genus Lactobacillus.

Out of these, 31.6% of the isolates were *Lb. paracasei*, 13.3% *Lb. helveticus*, 6.6% *Lb. amylolyticus*, 23.3% *Lb. paracasei* subsp. paracasei, 16.6% *Lb. plantarum*, 5%, *Lb. brevis*, 1.6% *Lb. coryniformis* and 16% *Lb. coryniformis* subsp. Torquens. *Lb. paracasei* were the most dominant species, having been detected in three samples, sample K, I and J, whilst three species appeared in two samples each, *Lb. plantarum*, (J and T), *Lb. paracasei* (K and I), *Lb. Helveticus* (D and I) as shown in (Appendix A). There were also two species that only appeared in one sample each, *Lb. amyloiticus* (D) and *Lb. brevis *(T). Sample J and I were the only samples with the highest number of isolates four and three, respectively. *Lb. brevis* was only recovered in sample (T). The evolutionary relationships among identified strains were inferred in a phylogenetic tree at 10,000 bootstraps using MEGA X, see Appendix A. Nine representative strains identified using 16S rRNA, were further subjected to phes gene sequencing to confirm their identity at the species level; isolate D12 (*Lb. amylolyticus*) was identified as *Lacticaseibacillus paracasei* subsp. Paracasei (Appendix A).

### 3.2. Antibacterial Activity of the Selected LABs against Pathogenic Microorganisms

Figure 1 shows antibacterial activity of LAB strains against *E. coli*, *S. aureus* and *S. typhimurium*. It was observed that all the tested LAB strains inhibited the indicator pathogen variably. Strain V (*Lactobacillus acidophilus*) and Y (*Lacticaseibacillus rhamnosus*) served as the positive control cultures with inhibition zones of zones of 37 mm and 40 mm, respectively. LAB strain T8 showed the highest levels of inhibition, with an inhibition zone of 42 mm, which is higher than the inhibition zone of the controls. Isolates K20, I12 and T12 exhibited inhibition zones ranging between 29 mm and 30 mm. The moderate inhibition zones (22 mm to 26 mm) were observed for J22, K5 and D7. The least inhibition ranging between 13.85 mm and 11.26 mm were observed for D12 and J20, respectively as depicted in Figure 1.

Regarding antibacterial activity against *S. aureus*, controls V and Y exhibited the highest inhibition zones ranging between 40 mm and 41 mm, respectively. The highest inhibition zone patterns were observed as D12 (33 mm) followed by K5 (32 mm), K20 (31 mm), I12 (30 mm), J22 (29 mm) and T12 (28 mm). The strains with the least inhibition were T8 (24 mm), D7 (16 mm) and J20 (10 mm) as shown in Figure 1. Similarly, the highest inhibition zones against *S. typhimurium* were observed in controls with the inhibition zones of 41 mm and 31 mm, for both *Lb. acidophilus* V and *Lb. rhamnosus* Y, respectively. Strain T8 showed a high level of inhibition, with an inhibition zone of 41 mm followed by I12 (36 mm). A lower degree of inhibition was observed for K20, D7, J22, K5 and T12, with inhibition zones ranging from 15.4 to 24.9 mm. The least inhibition zones were observed for strains J20 and D12, with inhibition zones of 6.11 and 1.57 mm, respectively, as shown in Figure 1.

### 3.3. Antibiotic Susceptibility Profiles of Isolated LABs

The results of the antibiotic resistance patterns of LAB strains tested in the current study are shown in Appendix A (Appendix A). All the nine tested LAB strains (K5, K20, J20, J22, T12, T8, D12, D7 and I12) were found to be susceptible to five antibiotics namely: ampicillin, erythromycin, mupirocin, tetracycline and chloramphenicol. These isolates were also observed to be resistant to polymyxin B, streptomycin, kanamycin and oxacillin. Strains J20 and T8 were susceptible to polymyxin B and streptomycin, while J22 was noted to be susceptible to polymyxin b and resistant to streptomycin.

### 3.4. Bile Tolerance Patterns of Isolated LABs

Figure 2A shows the tolerance pattern of selected LABs to 0.3% bile concentration under varying time intervals (initial stage, 5 and 24 h). A notable increase in biomass ranging between 6.3 log10 cfu/mL and 7.36 log cfu/mL for the control used at the initial stage and between 8.32 and 9.16 log10 cfu/mL at 24 h. Interestingly, selected LAB isolates were able to withstand 0.3% bile concentration over 24 h with an exception of D7, which was completely diminished within 5 h. Mean colony counts for the strain D7 was 5.35 log10 cfu/mL which drastically decreased after 5 and 24 h to 0 log10 cfu/mL for both time intervals, although the D7 strain increased exponentially to 8.15 log10 cfu/mL after 24 h. Strain T8 and T12 (18.18%) were the most tolerant to 0.3% bile with the mean colony counts increasing from 7.3 and 7.6 log10 cfu/mL at initial stage, to 8.9 and 9.2 log10 cfu/mL over 24 h, respectively. Strains K5, K20, J20, J22, D12, V and Y were the least tolerant to 0.3% bile. Generally, 2% bile salt yielded a higher growth inhibition degree compared to 0.3% bile for both controls and test isolates. A decrease (*p* < 0.05) in mean colony counts (from 5.15 log10 cfu/mL to 4.38 log10 cfu/mL) for all selected strains, with an exception of T8 (Lb. brevis) which showed an increase (*p* < 0.05) in colony counts (from 7.07 log10 cfu/mL to 7.46 log10 cfu/mL) after 5 h as shown in Figure 2B. Controls V and Y were also inhibited by the 2% bile salts at 5 h to 6-log10 cfu/mL after 24-h incubation. Strain D12 and T8 were able to survive the 2% bile concentration with mean counts of 8 log10 cfu/mL and 7.6 log10 cfu/mL, respectively in 24 h. Strains K5, K20 and T12 were the most sensitive to 2% bile. Mean colony counts decreased from 4.43 log10 cfu/mL to 3.15 log10 cfu/mL, 6.34 to 2.57 log10 cfu/mL and 3.69 to 2.73 Log10 cfu/mL, respectively, while D7 was completely inhibited from the initial stage to 24-h incubation as shown in Figure 2B.

### 3.5. Acid Tolerance Assays

As shown in Table 3, acid tolerance assays revealed an insignificant effect on the infallibility of most strains (*p* > 0.05) at pH 2 following an hour incubation. Only seven isolates lost viability at the end of the first hour. Similarly, the control strain (V) was also affected by pH 2 acidity stress in terms of viability. Control strain Y and two test strains, K20 and I12 were able to tolerate pH 2 at the 1 h interval and still had high counts of 5.97 log10 cfu/mL, (4.97 log10 cfu/mL) and Y (3.81 log10 cfu/mL), respectively. After 2 h, strain Y survived with 3.8 log10 cfu/mL compared to other strains that completely diminished at the same time interval.

Figure 3A (refer to Appendix A for standard errors) shows acid tolerance and viability of selected LABs at pH 3 under varying time intervals. After 1 h mean colony counts of five tested isolates which maintained their initial counts were, T8 (7.69–7.69 log10 cfu/mL), J22 (7.4–7.5 log10 cfu/mL). A slight decrease in mean colony counts were noted for control strains, V (6.69–6.59 log10 cfu/mL) and Y (7.7–7.64 log10 cfu/mL) in the first hour of being exposed to pH 3. Moreover, mean colony counts for strains J20 and K5 decreased from 7.5–6.87 and 7.23–6.67-log10 cfu/mL, respectively, in the first hour of exposure to pH 3. Only D7 and D12 had lost total viability after the first hour and 2 h, respectively. There was also a decrease in mean colony counts after 24 h of incubation. Isolates K5 decreased from 7.23–6.4 log 10 cfu/mL, K20 (7.09 to 6.89 log10 cfu/mL), J20 (7.5 to 6.33 log10 cfu/mL), J22 (7.4 to 6.65 log10 cfu/mL), T12 (8.33 to 7.09 log10 cfu/mL), (7.69 to 6.47 log10 cfu/mL) and I12 (7.22 to 6.63 log10 cfu/mL). There was also a slight decrease in the controls for both V (6.69 to 6.53 log10 cfu/mL) and Y (7.7 to 7.63 log10 cfu/mL). Figure 3B (refer to Appendix A for standard errors) depicts acid stress tolerance at pH 5. All selected LAB strains displayed a slight sensitivity to pH 5, as evidenced by average mean colony counts of 6.76 log10 cfu/mL at initial stage to 5.51 log10 cfu/mL after 24 h. However, the counts for the entire LAB cultures were maintained at a level higher that 6 log10 cfu/mL even after 24 h as shown in Figure 3B.

## 4. Discussion

Fermentation remains a preferred method of choice in food processing due to its ability to improve nutritional value of food products and palatability as well as consumer appeal [24,25]. Consequently, due to the increasing demand of gluten-free foods for individuals who have celiac diseases and wheat intolerance, sorghum has been recommended as a suitable alternative for these diet types [8]. Against this background, the current study identified and characterized LABs implicated in fermentation of a sorghum-fermented food product called (ting) and characterized the probiotic potential. Screening of LABs involved in fermentation of ting was achieved using a microbiological culture-based technique; following successive streaking, a total of 60 isolates presenting rod morphotypes on the surface of the media were initially identified as presumptive LABs. The isolated colonies were further subjected to phenotypic characterization and observed to be positive for both gram staining and oxidative test as shown in Table 2. The findings of the current study corroborate with previous research who observed isolated LABs from different food products (Mawe, Hussuwa, Uji, ogi and Kisra) to be positive for gram stain, oxidase and catalase-negative [26]. Typical LAB species are reported to be Gram-positive due to the physiological nature of their structure, hence they stain purple when subjected to a differential procedure such as Gram-staining [27,28]. This phenotypic characterization has helped other researchers to identify and confirm the identity of LABs without using laborious and costly molecular techniques. For example, [26] identified 25 out of 204 isolates from fermented food and beverages as LAB after being found to be Gram-positive, catalase-negative and rod-shaped [29] and [30] reported using the same methods. All isolates presenting rod-shaped morphology, Gram-positive and catalase-negative were further confirmed LABs isolated from sorghum and beverages to the genus level. It is well documented in the literature that Gram-positive bacteria are generally recognised as safe due to their use in various food applications [31]. These findings further affirm the relevance of microbiological culture-based and phenotypic characterization as an alternative cost mitigated approach for bioprospecting potential starter cultures towards improvement of fermentation technology.

Molecular analysis of the selected LABs revealed that the concentration and purity of the DNA was further quantified using the spectrophotometer (Nanodrop 2000) and showcased that the purity of all the DNA samples was between the expected purity levels of between 1.8 and 2 at 260/280 nm ratio, indicating a good quality of the extracted polymer. The size of the targeted gene was 1466 bp as presented in Appendix A. Other researchers have also confirmed the band size of the isolated DNA of the targeted LABs from various food applications to 443 kb in size [32]. The 16S rRNA gene is the commonly used gene for identification of LABs. However, this gene has its draw back as it cannot differentiate between closely related species, because the 16S rRNA evolves slowly, lacks sufficient diagnostic sites and has the 16S rDNA sequence similarity ranges from 90.9% to 99%. The use of protein coding genes such as the *pheS* gene has been used effectively to identify LAB species, as it can discriminate between closely related species [5,33]. Studies conducted by other researchers have suggested that in cases where isolates are ambiguous on the basis of 16S rRNA gene sequencing, the *pheS* and *rpoA* gene sequencing should be employed [5,22]. This result is similar to what other researchers have reported [5,22]. In this study the *pheS* genes were used to confirm the identification of the isolates at species levels. However, it was observed that strain D12 was erroneously identified using 16S rRNA. Nine Lactobacillus species were established as LABs present in household ting; *Lb. paracasei* was the most common and dominant organism (refer to Appendix A).

Although recovered isolates were somewhat similar, notable differences were observed for sample K and which were made of fine white sorghum flour and coarse white sorghum flour. Tested flour seems to possess its own set of unique LAB. Although there are similarities in some of the species across different flours, Sample K and I seem to be closely related in terms of the LAB identified. These findings iterate that, richness and diversity of LABs involved in fermentation of ting may differ owing to different formulations/flours used during preparation. The discrepancies associated with different formulations have also been observed by other researchers; for example, previous studies sought to isolate and identify different LABs found in sorghum slurries; *Lb. plantarum* and *Lb. paracasei* were found to be the most dominant species [7,30]. In contrast, the study conducted by [7] demonstrated the presence of unique LABs (*Lb. fermentum*, *Lb. parabucheri*, *Lb. reuteri* and *Lb. harbinensis*) which were not recovered in the current study. However, in terms of abundance, the present study isolated and recovered more Lactobaillus species (nine) in comparison to total number of studied samples [7,22,34]. To the best of the authors’ knowledge, this is the first study to isolate and document the implication of *Lb. helveticus* in sorghum-based fermented foods in Southern Africa. However, *Lb. helveticus* has been isolated by other researchers from different sources such as fermented kombucha and Kimchi [35]. In addition to formulation diversity, traditionally, sorghum fermentation is normally carried out at small- and household-scale. These are distinguished by the use of indigenous, non-sterile equipment and poor hygiene practices, which inevitably result in different microbial richness and diversity. This highlights the need for concerted efforts and research to aid in the formulation, standardization and improvement of the fermentation process. Furthermore, the inconsistencies of the ingredients and inoculum size may lead to varying pH in the final food products [8]; thus, effects of the physico-chemical characteristics of slurries should also be investigated due to their potential to have a significant effect on fermentation biological agents.

Among the criterion used to isolate microorganisms from fermented foods to function as probiotics, screening of antibiotic resistance and antimicrobial activity are of paramount importance to ascertain their potential health effects on consumers [36]. Thus, antibacterial activity of the nine LAB strains isolated in this study were tested on clinically important food pathogens namely *E. coli* ATCC 8739, *S. aureus* ATCC 6538 and *S. typhimurium* ATCC 14028. Overall, all nine tested LAB strains showed inhibitory effects against the three tested pathogens, although the degree of the inhibitory effects varied among the LAB strains in (Figure 1). The controls consistently showed good inhibition against the three pathogens with over 30 mm inhibition zones. Isolated LABs from this study were able to inhibit growth of selected pathogenic bacteria, presenting a highest inhibition zone against *E. coli* (11.26 to 42 mm) followed by *S. typhimurium* and *S. aureus*. Similarly, reported that the use of an overnight LAB culture had strong inhibitory activity against pathogenic bacteria, which corroborates with findings observed in the current study [37]. *E. coli* are gram-negative bacteria in the *Enterobacteriaceae* family that can colonize the human gut innocuously and cause intestinal or extra intestinal infections, including severe invasive disease such as bacteraemia and sepsis. *E. coli* is the most common cause of bacteraemia in both low and high-income countries, surpassing other pathogens that cause bacteraemia, such as *S. aureus* and *S. pneumoniae*, and is a leading cause of neonatal meningitis [38]. The results showed that although all nine strains had inhibitory effects against three strains, T8, K20 and I12 were consistent in the high level of inhibition against the three pathogens, which were either extremely strong, very strong or strong. Thus, in addition to well documented nutritional benefits associated with consumption of fermented sorghum food products, consumption of ting may also aid in therapeutic management of various clinical infections and foodborne diseases attributed to *E. coli*, *S. typhimurium* and *S. aureus*. Due to clinical complications caused by antibiotic resistance, there is a growing body of literature aimed at searching for alternative therapeutic agents, thus recovered LABs from this study are of paramount importance towards the development of probiotics functioning as alternatives to antibiotics, as evidenced by their ability to inhibit clinically important food pathogens.

Probiotic potential of fermented foods has been well documented and are generally recommended as safe due to their important properties in the pharmaceutical and food industry [17,39]. Previous studies have shown that they inhibit growth of pathogenic organisms through different mechanisms such as adherence to epithelial cells, modulation of the immune system, and secretion of antimicrobial compounds. The inhibitory substances are reported to be responsible for the inhibition of pathogens by secreting antimicrobial substances such as hydrogen peroxide, organic acids and bacteriocins [26,40,41,42]. Natural resistance to antibiotics can be a potential problem because it has been reported that Lactobacillus species enter the human gastro-intestinal tract in large numbers, where they interact with the intestinal microflora which may lead to transfer of resistance genes to pathogenic strains [43,44,45]. In this study, five selected antibiotics namely: Ampicillin, erythromycin, mupirocin, tetracycline and chloramphenicol were able to inhibit growth of all recovered LABs. Susceptibility to polymyxin B and streptomycin was also observed for T8 and J20. The antibiotic resistance and susceptibility patterns of the tested LAB strains were similar to the reference strains V and Y (Table 2) which were isolated from the commercial probiotic supplements by [5].

Since probiotics are administered orally, LAB strains have to survive the varying bile concentrations in the gastrointestinal tract and remain viable in high concentration (10^6^–10^8^ cfu/mL) to be considered as probiotics. It is reported that the liver releases about 1 L of bile, which decreases the survival of probiotic cells in the GIT [2,5].Consequently, studies have adopted 0.3% bile concentration as a standard for selection of bile tolerant probiotics [11,17]. In this study 90.9% of K5, K20, J20, J22, I12, T8 and T12 were able to survive 0.3% bile and remain viable in a high concentration record of 6 log10 cfu/mL as depicted in Figure 2A,B. The findings of the study are in line with other researchers [5,46,47] who have observed a survival growth rate of more than 6 log10 cfu/mL for *Lb. brevis* CE94, *Lb. brevis* CE85, *Lb. plantarum* CE42, *Lb. plantarum* CE60 and *Lb. plantarum* CE84 in 0.3% ox gall. In another study conducted by [37] it was observed that *Lb. acidophilus* was inhibited in 2% bile after 15 h although it still achieved 5 log10 cfu/mL colony counts. Therefore, LABs recovered from ting stand a higher chance of surviving fluctuating concentrations of bile salts in the intestinal passage and remain viable in high numbers. Therefore, these can be further investigated for their application as therapeutic potential probiotic strains. Tolerance to bile is attributed to the bile salt hydrolase enzyme which is reported to confer protection through bile conjugation, which catalyzes a reaction in which glycine and taurine are de-conjugated [48,49,50]. In addition to bile tolerance, pH is also considered as one of the important properties required for probiotic LABs to survive in the gastrointestinal tract.

Recommendations from previous research indicate that for *Lactobacillus strains* to be used as probiotics, it must be screened at pH 2 and pH 3 for not less than 120 min [5,47,51] in order to assess their potential to survive in the human gut. In the current study, recovered LABs were subjected to pH 2, pH 3 and pH 5 for a period of 24 h as shown in Table 3 and Figure 3A,B. As observed in this study, strains did not survive at pH 2 after 24 h of incubation. In this study, all the bacterial strains were very sensitive to acidic conditions at pH 2, as some strains did not survive at pH 2 under a stipulated period incubation of 1–3 h (Table 3). This finding implies that the tested strains will not retain viability or grow in the stomach at pH 2 for an extended period. Similar findings were observed in a study [52], which shows that pH 2 was the most stressing factor, as most of the isolates were not able to grow. In a study conducted by [53], it is reported that the gastric juices produced in the human stomach are a strong oxidizer which oxidizes important biomolecules in the cells such as fatty acids, proteins and cholesterol. Thus, in the human stomach, this has an effect on the viability of microorganisms, including potential probiotics due to the 2.5 L of gastric juice that is released in the stomach, which lowers the pH to 1.5 in the absence of food. However, in the presence of food, the pH can increase up to 3.5 [5,24,45]. The results in this study showed that six of the nine Lactobacillus strains exhibited good acid tolerance at pH 3 for 3 h, with three strains, J22, T12 and I12 showing significantly better acid tolerance as they remain viable until the end of the incubation period (Figure 3A). In order for the consumer to benefit from the consumption of probiotics, it is recommended that the strains maintain a therapeutic minimum inoculum of 10^6^–10^7^ cfu/mL in the gut [43,54]. Although most of them lost some viability, they still had viable mean colony counts above the recommended therapeutic minimum of 10^6^–10^7^ cfu/mL (Figure 3A,B). Generally, the results of the probiotic analyses obtained in this study show that seven out of nine LABs isolated from ting unequivocally possess the essential attributes of probiotics. As documented in previous studies, good candidacy criterion for probiotic isolates includes ability to withstand low pH, high resistance to bile salts in addition to offering health benefits such as antimicrobial activity and antibiotic resistance.

## 5. Conclusions

In this study, sixty LABs were established as unique morphotypes using microbiological culture-based and phenotypic characterization. Seven LABs thrive under low pH and high bile concentration in addition to possessing unparalleled antimicrobial activity against clinically important food pathogens, namely *E. coli* ATCC 8739, *S. Aureus* ATCC 6538 and *S. Typhimurium* ATCC 14,028 ranging between (1.57 to 41 mm), (10 to 41 mm) and (11.26 to 42 mm), respectively. Overall, the results of the probiotic potential analyses obtained in this study show that some LABs contained in ting unequivocally possess probiotics essential attributes. Further studies aimed at understanding the therapeutic role of such probiotics in the optimization of the gut microbiome may present a giant stride towards mitigation of the global neonatal mortality rate.

## Figures and Tables

**Figure 1 microorganisms-11-00715-f001:**
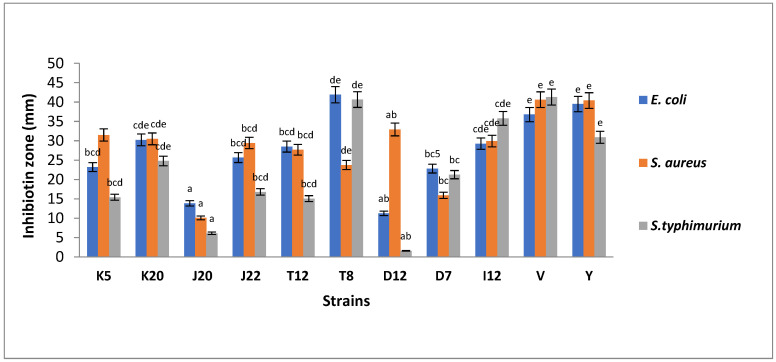
The antibacterial activity of LAB strains against pathogenic *E. coli* ATCC 8739, *S. aureus* ATCC 6538 and *S. typhimurium* ATCC 14028. Each bar represents mean of duplicate determinations (n = 2). Mean values with same letter/s are not significantly different (*p* > 0.05). Isolate codes (K5, K20, J20, J22, T12, T8, D12, D7 and I12) represented by single letter and number indicate the strain number where the strains were isolated.

**Figure 2 microorganisms-11-00715-f002:**
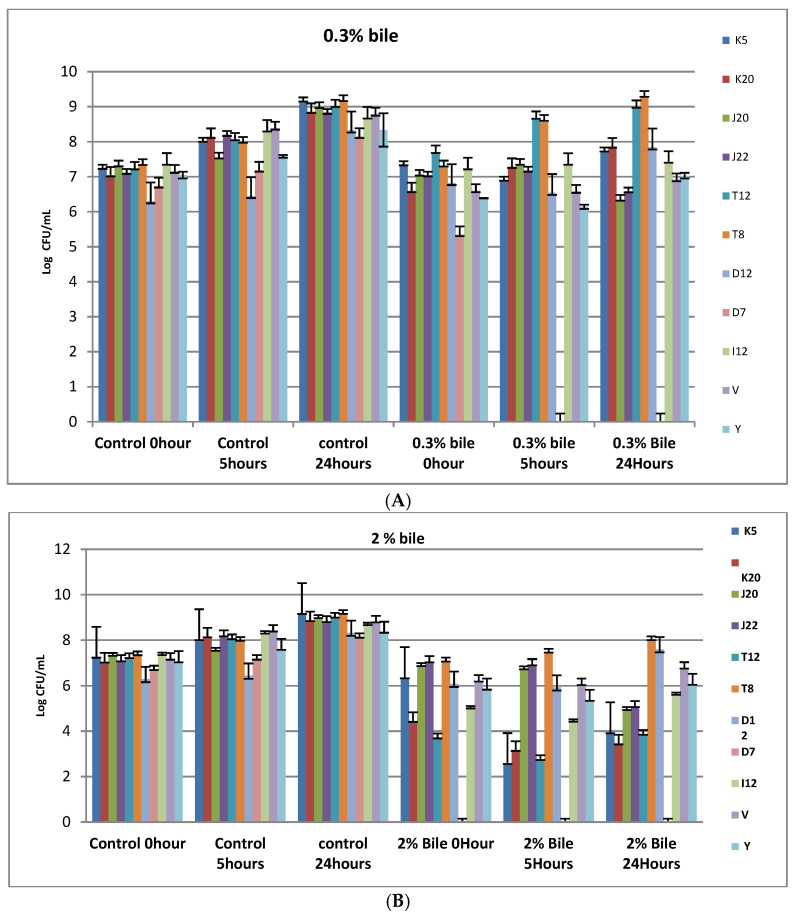
(**A**) Bile tolerance of LAB strains (K5, K20, J20, and J22, T12, T8, D12, D7, I12, V and Y) cultured in 0.3% bile and incubated at 37 °C for 24 h. Each bar represents the mean of duplicate determinations (n = 3); (**B**) Bile tolerance of LAB strains at 2% bile. Each bar represents the mean of duplicate determinations (n = 3). Isolate codes (K5, K20, J20, J22, T12, T8, D12, D7 and I12) represented by single letter and number indicate the strain number where the strains were isolated.

**Figure 3 microorganisms-11-00715-f003:**
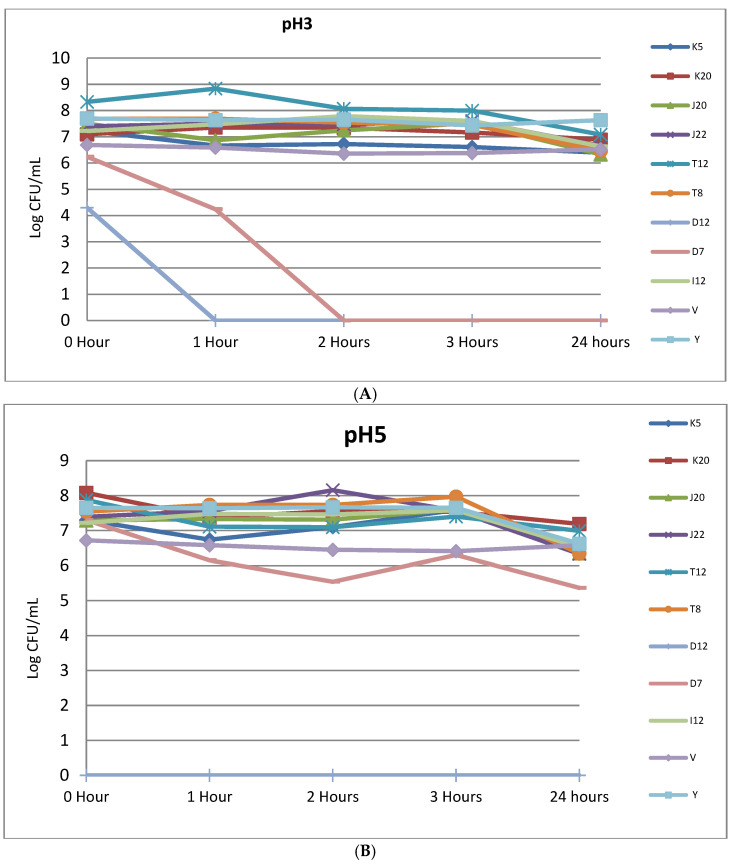
(**A**) Viability of LAB strains incubated in phosphate buffered saline at pH 3 for 24 h at 37 °C; (**B**) The viability of *Lactobacillus* strains subjected to acid-stress conditions at pH 5 for 24 h at 37 °C. Isolate codes (K5, K20, J20, J22, T12, T8, D12, D7 and I12) represented by single letter and number indicate the strain number where they were isolated.

**Table 1 microorganisms-11-00715-t001:** Primers used to amplify genes of *Lactobacillus* isolates.

Name	Nucleotide Sequence	Fragment Size	References
Phes 21-F	5′-CAYCCNGCHCGYGAYATGC-3′	411 bp	[23]
Phes-23-R	5′-GGRTGRACCATVCCNGCHCC-3′		
16S rRNA 27F 16S rRNA 1492R	5′-AGAGTTTGATCMTGGCTCAG-3′ 5′-GGTTACCTTGTTACGACTT-3′	1466 bp	[23]

**Table 2 microorganisms-11-00715-t002:** Phenotypic characterization of sorghum isolates from spontaneously fermented sorghum slurries.

Scheme	No of Isolates	Isolate Codes	Morphology	Grams Reaction	Catalase Reaction	Oxidase Reaction
D—Coarse sorghum from North of Pretoria	11	D5, D13, D16, D18, D7, D19, D2, D3, D11, D12, D20	Rods	+	−	−
K—Fine sorghum from Klipgat	17	K5, K20, K19, K18, K11, K10, K9, K8, K3 K1, K2, K17, K16, K12, K7, K6, K4	Rods	+	−	−
I—coarse sorghum from Soshanguve	15	I6, I5, I7, I10, I21, I17, I16, I13, I1, I2, I15, I12, I11, I19, I20	Rods	+	−	−
J—Coarse White sorghum from Pretoria	9	J21, J6, J14, J15, J16, J19, J7, J22, J20	Rods	+	−	−
T—Fine Brown sorghum from Tembisa	8	T2, T5, T10, T12, T20 T6, T8, T16	Rods	+	−	−

Sample ID (D, I, J, K and T) represented by a single letter indicating the place where the samples were taken.

**Table 3 microorganisms-11-00715-t003:** Viability (log CFU/mL) of LAB strains incubated in phosphate buffered saline at pH 2 at 37 °C.

Strains	Initial Concentration	1 h	2 h	3 h	24 h
K5	7.06 ± 0.19 ^a^	0	0	0	0
K20	7.13 ± 0.03 ^a^	4.97 ± 0.09 ^a^	0	0	0
J20	1.72 ± 0.13 ^a^	0	0	0	0
J22	7.14 ± 0.04 ^a^	0	0	0	0
T12	7.84 ± 0.14 ^a^	0	0	0	0
T8	7.54 ± 0.09 ^a^	0	0	0	0
D12	6.3 ± 0.05 ^a^	0	0	0	0
D7	6.8 ± 0.014 ^a^	0	0	0	0
I12	7.3 ± 0.09 ^a^	5.97 ± 0.24 ^a^	0	0	0
V	6.21 ± 0.12 ^a^	0	0	0	0
Y	7.9 ± 0.16 ^a^	3.65 ± 0.38 ^a^	3.81 ± 0.15 ^a^	0	0

Mean values with same letter/s in the same columns is not significantly different (*p* > 0.05). Isolate codes (K5, K20, J20, J22, T12, T8, D12, D7 and I12) represented by single letter and number indicate the strain number where they were isolated.

## Data Availability

All relevant data generated in this study are available in this manuscript and a Appendix A.

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
