# Peer review of "Recovery of Potential Starter Cultures and Probiotics from Fermented Sorghum (Ting) Slurries"

_microorganisms, 2023, doi:10.3390/microorganisms11030715_

Round 1
Reviewer 1 Report
please correct the highlighted sentences in the attached article and clear it
the abstract is very short and needs more explanation
reference, figures, and tables should be written according to the journal format
abbreviations should be written in completely at first and should be cleared
Each statement should be a reference
The primers used in the material and method preferred to illustrate in the table
What is the number of samples used for the sequence?
Put the results of morphological and biochemical in isolated paragraph
The number of isolates 60 while mentioned in the figure from 1-29 needs correction
The six feed samples examined take a letter in result should be cleared in material and method
The feed sample letters and isolate numbers should be cleared all over the article
The result should be explained and cleared

Author Response
Reviewer 1:
Point 1: Please correct the highlighted sentences in the attached article and clear it
the abstract is very short and needs more explanation
Response: The said sentences corrected as per reviewer’s comments, please see line 8 – 10. The abstract was initially shortened according to the number of words required during submission.
Point 2: Reference, figures, and tables should be written according to the journal format abbreviations should be written in completely at first and should be cleared. Each statement should be a reference.
Response: MDPI reference adopted throughout the manuscript.
Point 3: The primers used in the material and method preferred to illustrate in the table-
Response: Table inserted- line 129.
Point 4: What is the number of samples used for the sequence?
Response: 60 PCR products- line number 132.
Point 5: Put the results of morphological and biochemical in isolated paragraph.
Response: The results for morphological and biochemical tests were isolated and written in one paragraph. Line number 186 – 191.
Point 6: The number of isolates 60 while mentioned in the figure from 1-29 needs correction.
Response: There are three figures in the supplementary data. The loading wells can only run 29 samples at a time and the first loading well is the marker M. The last figure depicts the remaining two DNA samples that adds up to in total.
Point 7: The six feed samples examined take a letter in result should be cleared in material and method
Response: *Sample ID (D, I, J, K and T) represented by single letter indicate the place where the samples were taken. It is already explained in the manuscript; please see line 191-192.
Point 8: The feed sample letters and isolate numbers should be cleared all over the article
Response: Sample ID (D, I, J, K and T) represented by single letter indicate the place where the samples were taken. It is clarified in the manuscript. Please see line 191-192.
Point 9: The result should be explained and cleared.
Response: The results are explained in the main manuscript.
Reviewer 2 Report
Peer-review report of the research article (microorganisms-2177454)
The manuscript entitled, “Recovery of potential starter cultures and probiotics from fermented sorghum (Ting) slurries” is an excellent piece of research submitted for publication in the journal “microorganisms.”
This manuscript needs a major revision before acceptance.
The isolation of new species having therapeutic effects on human health attracts scientists and related groups. Moreover, studying fermented foods is a hot topic due to their varied flavors and health-alleviating effects. Keeping in view the importance of this topic, the review report of the aforementioned article is as follows.
Overall assessment: The idea of this article is well-conceived and workable. The researchers have planned and conducted the experiments in a better way. There are some discrepancies, which need to be addressed.
The species from the Lactobacillus genus are selected for the study. Why are other genera not included in this study?
The logic behind the selection of sites for sample selection is missing.
Variation of significance level in the formulation of “Ting” is missing.
The significance of the species isolated in the present study is incomplete. These species are previously reported, and the authors should elaborate on the versatility of these species in various foods, such as other fermented foods, especially those used commercially. If possible, they should include the best possible benefit of the isolated species.
Abstract: The abstract is well-written presenting a good summary of the whole manuscript.
It is mentioned, “…in 7 out of 9 under low pH and high bile concentration in vitro.” Mention the exact pH and bile concentration.
It is mentioned, “…for S. typhimurium, S. aureus and E. coli, respectively.” Mention the exact names of the microorganisms, including strain numbers, etc.
Introduction: The article is structured to fulfill the requirements of a research article. The introduction is written precisely.
In the introduction, mention a little history of “Ting”.
Its origin and start of using.
Brief method and season of fermentation.
Material and Methods: The material and methods are well-written and properly cited.
Explain the logic behind the selection of 0, 5, and 24 hours for the bile tolerance test.
Results and discussion:
It is mentioned, “…there were also 3 species that only appeared in one sample each, Lb. amyloiticus (D) and Levilactobacillus brevis (T).” Which one is the third species?
It is mentioned, “Sample J and I were the most abundant (4) each.” The meaning is not clear.
It is mentioned, “Strain V and Y served as the positive control cultures…” What are name of strain V and Y?
It is mentioned, “All the nine tested LAB strains (K5, K20, J20, J22, T12, T8, D12, D7 and 12) were…” Is there any strain named “12” in this experiment?
Table 2. What is meaning of “0 hours?” If it is the start of the experiment, why there is a variation in the initial concentration of the inoculums?
It is mentioned, “…isolated LABs from different food products to be positive for both gram stain and catalase.” Name some foods, especially close to “Ting”.
It is mentioned, “…Studies conducted by has suggested that in the case were isolates are ambiguous…” Who conducted the study?
It is mentioned, “The discrepancies associated with……… to be the most dominant species.” Is “Ting” not a sorghum slurry? In introduction, the authors mentioned that no study regarding sorghum slurries is conducted, and here they are giving references to the previous studies.
It is mentioned, “However, in terms of abundance, the present study isolated and recovered more Lactobaillus species (9) in comparison to …” Is this the highest number of isolated LAB species from a sample?
It is mentioned, “These are distinguished by the use of indigenous, non-sterile equipment and poor hygiene practice, which inevitably result in different microbial diversity and nutritional benefits.” The notion implies that non-sterile equipment and poor hygiene are good to produce versatility and improve nutrition. Is this true?
It is mentioned, “Similarly, (Pan et al., 2009) reported that the use of overnight LAB culture…” Is in-text citation used properly?
It is mentioned, “…strains were very sensitive to acidic conditions at pH 2 as did not survive at pH 2 under a stipulated period of incubation…” Mention the “stipulated period.”
It is mentioned, “…with three strains showing significantly better acid tolerance…” Name the three strains.
It is mentioned, “…it is recommended that the strains maintain a therapeutic minimum inoculum of 106 - 107 cfu/mL.” How it is possible to maintain this concentration is a household slurry?
Conclusion: The conclusion is written properly; however, it must demonstrate the most specific findings of the study, and not all results.
Formatting: The manuscript must be thoroughly formatted as several inconsistencies are found throughout the manuscript.
The name of bacterial strains are inconsistent throughout the manuscript.
Use of space is inconsistent.
The in-text citation use is inconsistent.
The quality of images is poor. Improve the images in terms of data presentation and readability.
There is no “Table 3” in the manuscript; however, references to “Table 3” are present in the “Discussion” section.
Language: The language and grammar of the manuscript must be improved.
Author Response
Responses to Reviewer 2
Point 1:
Peer-review report of the research article (microorganisms-2177454)
The manuscript entitled, “Recovery of potential starter cultures and probiotics from fermented sorghum (Ting) slurries” is an excellent piece of research submitted for publication in the journal “microorganisms.” This manuscript needs a major revision before acceptance. The isolation of new species having therapeutic effects on human health attracts scientists and related groups. Moreover, studying fermented foods is a hot topic due to their varied flavors and health-alleviating effects. Keeping in view the importance of this topic, the review report of the aforementioned article is as follows.
Overall assessment: The idea of this article is well conceived and workable. The researchers have planned and conducted the experiments in a better way. There are some discrepancies, which need to be addressed.
Point 2: The species from the Lactobacillus genus are selected for the study. Why are other genera not included in this study?
Response: We only worked with what we managed to isolate from the sorghum slurries (ting) and later tested for their probiotic potential and controls were used which were previously isolated and characterized by Nyanzi (2013).
Point 3: The logic behind the selection of sites for sample selection is missing.
Response: Sample collection was clearly explained and incorporated, from Line 87-93 in section 2.1
Point 4: The significance of the species isolated in the present study is incomplete. These species are previously reported, and the authors should elaborate on the versatility of these species in various foods, such as other fermented foods, especially those used commercially. If possible, they should include the best possible benefit of the isolated species.
Response: The table recording the recovered isolates and their functions has been added, please see Table S.
Point 5: The abstract is well-written presenting a good summary of the whole manuscript.It is mentioned, “…in 7 out of 9 under low pH and high bile concentration in vitro.” Mention the exact pH and bile concentration.
Response: The exact pH and Bile concentration were included- line number 16-17
Point 6: It is mentioned, “…for S. typhimurium, S. aureus and E. coli, respectively.” Mention the exact names of the microorganisms, including strain numbers, etc.
Response: The names of the microorganisms were written in full with strains numbers. Line 19-20.
Point 7: The article is structured to fulfil the requirements of a research article. The introduction is written precisely. In the introduction, mention a little history of “Ting”. Its origin and start of using. Brief method and season of fermentation.
Response: History of ting, its origin and method and season addressed in the introduction-line 44-49.
Point 8: The material and methods are well-written and properly cited. Explain the logic behind the selection of 0, 5, and 24 hours for the bile tolerance test.
Response: Due to the bile salt excreted from the liver, bile concentration in the GIT fluctuates in the range 1.5% - 2.0% (w/v) during the first hour of digestion and subsides to 0.3% (Bao et al., 2010). We wanted to determine their survival rate from start of consumption as it is a requirement for LABSs to survive in high number (106) in order for one to benefit from the health effects they confer to human health. Hence we selected 0, 5 hours and 24hours.
Point 9: It is mentioned, “…there were also 3 species that only appeared in one sample each, Lb.amyloiticus (D) and Levilactobacillus brevis (T).” Which one is the third species?
Response: Typing error is 2 species. Line 206.
Point 10: It is mentioned, “Sample J and I were the most abundant (4) each.” The meaning is not clear.
Response: Sample J and I were only samples with the highest number of isolates 4 and 3, respectively. Line 208-209
Point 11: It is mentioned, “Strain V and Y served as the positive control cultures…” What are name ofstrain V and Y?
Response: Strain V (Lb. acidophilus) and Y (Lb. rhamnosus)- line 219.
Point 12: It is mentioned, “All the nine tested LAB strains (K5, K20, J20, J22, T12, T8, D12, D7 and 12) were…” Is there any strain named “12” in this experiment?
Response: Typing error addressed- line 245-246.
Point 13: Table 2. What is meaning of “0 hours?” If it is the start of the experiment, why there is a variationin the initial concentration of the inoculums?
Response: 0 hour was replaced with (initial concentration). About 1ml of inoculum was used for all tested isolates and their counts were determined at initial stage- line 254-313.
Point 14: It is mentioned, “…isolated LABs from different food products to be positive for both gram stain and catalase.” Name some foods, especially close to “Ting”.
Response: Mawe, Hussuwa, Uji, ogi, Kisra, changes addressed. Line 325.
Point 15: It is mentioned, “…Studies conducted by has suggested that in the case were isolates are ambiguous…” Who conducted the study?
Response: (Nyanzi 2013 and Madoroba et al., 2011). Line 354-355.
Point 16: It is mentioned, “The discrepancies associated with……… to be the most dominant species.” Is“Ting” not a sorghum slurry? In introduction, the authors mentioned that no study regarding sorghum slurries is conducted, and here they are giving references to the previous studies.
Response: Studies on sorghum slurries have been carried out before, however; they focused mainly on physico-chemical characterization of sorghum, richness and diversity of lactic acid bacteria without screening for probiotic potential. This study records the earliest record of probiotic potential strains.
Point 17: It is mentioned, “However, in terms of abundance, the present study isolated and recovered more Lactobaillus species (9) in comparison to …” Is this the highest number of isolated LAB species from a sample?
Response: Is the highest number of isolates in our total number of collected samples in comparison to studies conducted by Sekwati-bonang, and Ganzle, 2011; Madoroba, 2011). Line 380-383.
Point 16: It is mentioned, “These are distinguished by the use of indigenous, non-sterile equipment and poor hygiene practice, which inevitably result in different microbial diversity and nutritional benefits.” The notion implies that non-sterile equipment and poor hygiene are good to produce versatility and improve nutrition. Is this true?
Response: The sentence reframed and corrected. Please see line 397
Point 17: It is mentioned, “Similarly, (Pan et al., 2009) reported that the use of overnight LAB culture…” Isin-text citation used properly?.
Response: The in text citation has been corrected…. Please see line 407
Point 18: It is mentioned, “…strains were very sensitive to acidic conditions at pH 2 as did not survive atpH 2 under a stipulated period of incubation…” Mention the “stipulated period.”
Response: Stipulated incubation period is 1-3 hours. Line 473.
Point 19: It is mentioned, “…with three strains showing significantly better acid tolerance…” Name the three strains.
Response: Strains J22, T12 and I12 changes named. Line 485.
Point 20: It is mentioned, “…it is recommended that the strains maintain a therapeutic minimum inoculumof 10 - 10 cfu/mL.” How it is possible to maintain this concentration is a household slurry?
Response: The sentence clarified. Please see line 495.
Point 19: The conclusion is written properly; however, it must demonstrate the most specific findings of the study, and not all results.
Response: The conclusion shortened as per reviwer’s. Please line 505 – 515.
Point 20: The manuscript must be thoroughly formatted as several inconsistencies are found throughout the manuscript.
Response: The inconsistencies fixed throughout the manuscript.
Point 21: The name of bacterial strains are inconsistent throughout the manuscript.
Use of space is inconsistent. The in-text citation use is inconsistent.
Response: Inconsistencies in-texts reference were amended.
Point 22: The quality of images is poor. Improve the images in terms of data presentation and readability.
Response: The figure qualities have been improved throughout the manuscript.
Point 22: There is no “Table 3” in the manuscript; however, references to “Table 3” are present in the“Discussion” section.
Response: Typing error fixed, it was supposed to be Table 3-line 290.
Reviewer 3 Report
Author should include line number throughout the article that will help in review process.
Line-3 author had written .. “beneficial in alleviation of metabolic diseases”- if it enhance the metabolic diseses than how it is beneficial.
- Check the sentence: l foods inevitably driven the heightened consumption of sorghum-based food products such as Ting
- Reframe the sentence: However, increasing body of literature citing bacterial resistance against commonly prescribed antibiotics have sparked a public concern, with some calling for their discontinuation
- Please add citation in “2.2. Molecular characterisation of selected LABs”
- Check 2.3.”16. S rRNA “and pheS sequencing and phylogenetic analysis- Remove dot after 16.
- No relevancy of table.1, it can be added in the text, because no any differences among them
- Increase the text size of Fig.2 specially bar
- Strictly follow MDPI reference style
Author Response
Reviewer 3
Point 1: Line-3 author had written.. “beneficial in alleviation of metabolic diseases”- if it enhance the metabolic diseases than how it is beneficial.
Response: The word ‘alleviate’ means to lessen/mitigate, please double check.
Point 2: Check the sentence: l foods inevitably driven the heightened consumption of sorghum-based food products such as Ting
Response: Thank you pointin that out, the sentence has been fixed. Please see line 495.
Point 3: Reframe the sentence: However, increasing body of literature citing bacterial resistance against commonly prescribed antibiotics have sparked a public concern, with some calling for their discontinuation
Response: The sentence corrected, see line 69 – 70.
Point 4: Please add citation in “2.2. Molecular characterisation of selected LABs”.
Response: Selected LAB isolates were characterised at molecular level using 16S rRNA and pheynylalanyl tRNA synthetase ulpha subunit (pheS) gene sequencing by following method described by Nyanzi (2013). See line 105-107.
Point 5: Check 2.3.”16. S rRNA “and pheS sequencing and phylogenetic analysis- Remove dot after 16.
Response: As pointed out by the reviewer, the dot was removed after 16. See Line 131.
Point 6: No relevancy of table.1, it can be added in the text, because no any differences among them.
Response: Table was inserted in the texts. See Line 191.
Point 7: Increase the text size of Fig.2 specially bar.
Response: The text size increased. See line 239-244.
Point 8: Strictly follow MDPI reference style
Response: MDPI reference style adopted throughout the manuscript
Round 2
Reviewer 3 Report
Authors have significant improved the article. Article can be accepted in the present form